# Administration of 3% Sodium Chloride and Local Infusion Reactions

**DOI:** 10.3390/children9081245

**Published:** 2022-08-18

**Authors:** Michael L. Moritz, Juan C. Ayus, Joel B. Nelson

**Affiliations:** 1Division of Nephrology, Department of Pediatrics, University of Pittsburgh School of Medicine, Pittsburgh, PA 15213, USA; 2Division of Nephrology, UPMC Children’s Hospital of Pittsburgh, Pittsburgh, PA 15224, USA; 3Division of Nephrology and Hypertension and Kidney Transplantation, University of California Irvine, Orange, CA 92617, USA; 4Department of Urology, University of Pittsburgh School of Medicine, Pittsburgh, PA 15213, USA

**Keywords:** hypertonic saline, intravenous, infiltration, extravasation, hyponatremia

## Abstract

Three-percent sodium chloride (3% NaCl) is a hyperosmolar agent used to treat hyponatremic encephalopathy or other cases of increased intracranial pressure. A barrier to the use of 3% NaCl is the perceived risk of local infusion reactions when administered through a peripheral vein. We sought to evaluate reports of local infusion reactions associated with 3% NaCl over a 10-year period throughout a large healthcare system. A query was conducted through the Risk Master database to determine if there were any local infusion reactions associated with peripheral 3% NaCl administration throughout the entire UPMC health system, which consists of 40 hospitals with 8400 licensed beds, over a 10-year time period from 14 May 2010 to 14 May 2020. Search terms included infiltrations, extravasations, phlebitis, IV site issues, and IV solutions. There were 23,714 non-chemotherapeutic and non-contrast-associated intravenous events, of which 4678 (19.7%) were at UPMC Children’s Hospital. A total of 2306 patients received 3% NaCl, of whom 836 (35.8%) were at UPMC Children’s Hospital. There were no reported local infusion reactions with 3% NaCl. There were no reported local infusion reaction events associated with 3% NaCl in a large healthcare system over a 10-year period. This suggests that 3% NaCl can be safely administered through a peripheral IV or central venous catheter.

## 1. Introduction

Three-percent sodium chloride (3% NaCl, Na 513 mEq/L, 1027 mOsm/L) is a hyperosmolar agent primarily indicated for the treatment of hyponatremic encephalopathy or to raise the serum osmolality in other cases of increased intracranial pressure [1,2]. In addition, 3% NaCl can be useful in asymptomatic forms of hyponatremia, such as those that stem from the syndrome of inappropriate antidiuresis or congestive heart failure, in which conservative measures such as fluid restriction and 0.9% percent sodium chloride are frequently ineffective [3,4]. Furthermore, 3% NaCl is sufficiently hypertonic to raise the serum sodium in any form of hyponatremia. A barrier to the use of 3% NaCl is the perceived risk of local infusion reactions when administered through a peripheral vein [5,6]. Placement of a central venous catheter (CVC) to administer 3% NaCl could delay potentially lifesaving therapy, cause discomfort to patients, and result in complications, such as bloodstream infections, venous thrombosis, and stenosis. Local infusion reactions through a peripheral vein are known to occur with high-concentration total parenteral nutrition, potassium, calcium, and 24% NaCl, though there have been few reports of serious complications with the peripheral administration of 3% NaCl [7,8]. A survey we conducted of a network of children’s hospital pharmacies found that 57% had restrictions on administering 3% NaCl through a peripheral vein [9]. We conducted a systematic literature review and found infusion-related adverse events from the peripheral administration of 3% NaCl to be uncommon [10]. A quality improved project was conducted to determine if there were any reported events of infiltrations or extravasations associated with 3% NaCl administration throughout the UPMC system, as UPMC does allow 3% NaCl to be administered through a peripheral vein.

## 2. Methods

A query was conducted through the Risk Master database to determine if there were infiltrations or extravasations associated with hypertonic saline administration throughout the UPMC health system during a 10-year time period from 14 May 2010 to 14 May 2020 (Figure 1). The search was UPMC system-wide, including all acute care facilities, home care services, senior living facilities, rehab centers, cancer centers, personal care settings, physician offices, and ambulatory care centers. The UPMC health system consists of 40 hospitals with 8400 licensed beds. The query involved both adults and pediatrics.

## 3. Results

During that timeframe, 2306 patients were administered 3% NaCl, of whom 826 (35.8%) were at UPMC Children’s Hospital. Data were not available about the route of administration, i.e., central or peripheral administration.

There were 1,153,099 Risk Master events reported in that time frame, of which 29,573 (2.56%) were closely associated with six categories of IV extravasation: IV infiltration, IV site issues, IV phlebitis, IV solution, X-ray contrast infiltration, and chemotherapy infiltrations. A total of 23,714 of these events (82%) were non-chemotherapeutic or non-contrast events, of which 617 (2.59%) were deemed serious events by a patient safety officer. Of the reported events associated with IV extravasation, 4648 (19.7%) were at UPMC Children’s Hospital.

A separate search was then conducted with the term hypertonic in the event description, which resulted in 96 hypertonic events. Of the 24,317 IV-related extravasation events, 6 (0.02%) had the word hypertonic in the description, with three events related to infiltration. Of these, one was unrelated to fluid administration, one was related to 0.9% NaCl, and one was related to 1.5% NaCl. All were mild and resulted in the removal of a 20 g IV. There were no reported infiltrations or extravasations with 3% NaCl or hypertonic sodium bicarbonate.

## 4. Discussion

This study demonstrates that there were no reports of infiltration or extravasation associated with 3% NaCl being administered in the entire UPMC health system, in both pediatrics and adults, over a 10-year period. UPMC Children’s Hospital accounted for a large proportion of patients receiving 3% NaCl and with IV events. To the best of our knowledge, this is the largest search of a health system related to complications of intravenous extravasations associated with 3% NaCl administration.

Our findings are consistent with those of others. Single-center retrospective studies have failed to demonstrate an association between the peripheral administration of 3% NaCl and local infusion reactions [7,8,11]. Similarly, local infusion reactions were infrequent in three prospective studies involving 289 adults who received 3% NaCl through a peripheral vein to treat hyponatremic encephalopathy [12,13,14]. Clinical practice guidelines recommend prompt administration of 3% NaCl to treat symptomatic hyponatremia. Placement of a CVC could lead to a potentially dangerous delay in therapy in an emergency [13,15]. Many children’s hospitals have pharmacy restrictions regarding the administration of 3% NaCl through a peripheral vein [9]. The data from this study supports the administration of 3% NaCl through a peripheral vein.

This query was unable to ascertain the route of administration of 3% NaCl, i.e., through a peripheral vein or central venous catheter. It was also unable to evaluate the number of doses, rate, quantity, and duration of 3% NaCl administration. UPMC pharmacy policy does allow 3% NaCl to be administered through a peripheral IV, and it is routinely administered in head trauma patients, so many doses of 3% NaCl were likely administered through a peripheral IV. This suggests that local complications related to the peripheral administration of 3% NaCl are uncommon and that the peripheral administration of 3% NaCl is infrequently associated with local infusion reactions. It is possible that complications related to IV extravasation went unreported. Of the almost 25,000 IV events reported, however, none were related to 3% NaCl. This report suggests that 3% NaCl can likely be safely administered through a peripheral IV if a CVC is not available.

This study did not assess other complications related to 3% NaCl administration, namely cerebral demyelination related to overcorrection of hyponatremia. Animal studies and case reports in humans have demonstrated that overcorrection of severe (<115 mEq/L) and chronic hyponatremia (>48 h) by >25 mEq/L can produce brain injury from cerebral demyelination [16,17]. There are numerous risk factors for developing demyelination, independent of the rate of correction, including severe liver disease, alcoholism, hypokalemia, hypophosphatemia, hypoxia, and malnutrition [18]. When demyelination occurs in high-risk patients, it can be difficult to ascertain what role the correction of hyponatremia played. For these reasons, there is controversy related to safe limits for correction, and expert recommendations vary [15]. We have proposed using intermittent 3% NaCl boluses in order to achieve an acute elevation in serum sodium while at the same time avoiding overcorrection with a prolonged 3% NaCl infusion [19]. This approach has been incorporated into the European Clinical Practice Guidelines [15]. Three recent studies compared 3% NaCl bolus with continuous infusions or conventional therapy, and all found that the increase in serum sodium was more rapid and consistent with the bolus approach and did not contribute to demyelination [13,14,20]. Overcorrection was encountered in both approaches, was primarily related to the severity of hyponatremia and a free water diuresis, and was not associated with demyelination. Two of the studies were prospective and specifically reported on peripheral vein use in 225 patients without reported infusion reactions [13,14]. Intermittent bolus therapy of 3% NaCl may be able to improve safety, though overcorrection can occur regardless of which therapy is used.

In conclusion, this study found no reports of local infusion reactions associated with 3% NaCl administration over a 10-year-period in a large healthcare system. The major limitation of this study was that the route of administration (peripheral vein or central venous catheter) could not be assessed. The findings of this study are consistent with those of other retrospective and prospective studies in that local infusion reactions were uncommon with 3% NaCl administration through a peripheral vein. Potentially life-saving therapy with 3% NaCl should not be delayed in order to place a central venous catheter.

## Figures and Tables

**Figure 1 children-09-01245-f001:**
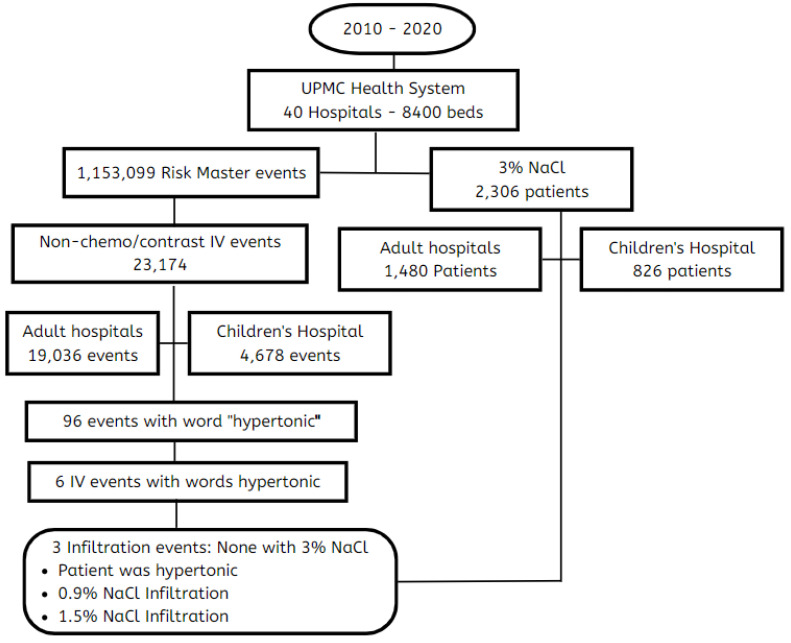
Health system-wide query of local intravenous reactions association with 3% NaCl. No reported infiltration reactions associated with 3% NaCl throughout the UPMC Health System in a 10-year period.

## Data Availability

This is not relevant for this type of data. All relevant data is in the manuscript.

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
