# Peer review of "Administration of 3% Sodium Chloride and Local Infusion Reactions"

_children, 2022, doi:10.3390/children9081245_

Round 1

Reviewer 1 Report

Reviewer comments

Thank you for the opportunity of reviewing your great and interesting work.

In real clinical world, the physicians frequently encounter the concerns of routes or doses of 3% NaCl uses. This is a good article about this topic, I think.

However, as the authors already had written the limitation of the works, it is the huge limitation of unavailability of access of the routes, peripheral or CVC. Nevertheless the authors concluded that the infusion of 3% NaCl through peripheral IV is safe. I think this can be improper conclusion. At least, you should write the safe infusion through peripheral or central vein is safe, if you cannot distinguish the routes of infusion.

One more thing, you should express the numbers of pediatric and adult patients. The indications of 3% NaCl infusion are likely to be applied to the adult patients. Therefore if the readers can notice the numbers of the patients, its clinical significances can be recognized certainly.

Author Response

Thank you for those extremely helpful comments. Below is our point by point response

Reviewer:

"However, as the authors already had written the limitation of the works, it is the huge limitation of unavailability of access of the routes, peripheral or CVC. Nevertheless the authors concluded that the infusion of 3% NaCl through peripheral IV is safe. I think this can be improper conclusion. At least, you should write the safe infusion through peripheral or central vein is safe, if you cannot distinguish the routes of infusion."

Response: 

We have now added to the conclusion of the abstract that “This suggests that 3% NaCl can be safely administered through a peripheral IV or central venous catheter.”

Reviewer: 

"...you should express the numbers of pediatric and adult patients."

Response:

Thank you for requesting the pediatric data on 3% NaCl administration.  When we repeated the data analysis it became apparent that it had inadvertently not captured UPMC Children’s Hospital as they are on different EMR systems.  We pulled the data from UPMC Children’s and there were many pediatric cases. We have modified the abstract, figure and results to reflect this.

Reviewer 2 Report

Manuscript entitled “Administration of 3% sodium chloride and local infusion reactions” (Moritz ML et al.)

Metheny NA and Moritz ML recently performed a systematic review of the literature and observed that in patients who receive 3% NaCl through a peripheral vein, infusion reactions are either uncommon or no more frequent than with routine solutions (Metheny NA, Moritz ML. Administration of 3% Sodium Chloride via a peripheral vein. J Infus Nurs 2021;44(2):94-102).

The same group conducted a quality improved projec to determine if there are any reported events of infiltrations or extravasations associated with 3% NaCl administration throughout the UPMC system. Again, there were no reported local infusion reactions with 3% NaCl over a 10-year-period. The authors concluded that that 3% NaCl can be safely administered through a peripheral vein.

The report is well written, interesting a rather concise.

Minor comments:

- Abstract (penultimate sentence sentence): “over a 10-year-perios”

- Generally, the length of the manuscript might well be reduced by ≤10%

- I wonder if the literature is presented as recommended by the journal “Children”

Author Response

Thank you for the comments. Below is point by point response.

Reviewer: 

Abstract (penultimate sentence sentence): “over a 10-year-perios

Response:

Thank you for noticing this. It was corrected.

Reviewer:

Generally, the length of the manuscript might well be reduced by ≤10%

Response:

We agree that the manuscript could be shorter. When we submitted the original manuscript, the editor believed that the manuscript was too short and they requested that we increase the length of the manuscript. We therefore expanded the discussion at the request of the editor.